# Global patterns in price elasticities of sugar-sweetened beverage intake and potential effectiveness of tax policy: a cross-sectional study of 164 countries by sex, age and global-income decile

Andrew Muhammad,[1] Birgit Meade,[2] David R Marquardt,[2] Dariush Mozaffarian[3]

[1]University of Tennessee Institute of Agriculture, Knoxville, Tennessee, USA
[2]USDA Economic Research Service, Washington, DC, USA
[3]Tufts University Friedman School of Nutrition Science and Policy, Boston, Massachusetts, USA

**Correspondence to**
Dr Andrew Muhammad;
amuhammad@utk.edu

## ABSTRACT

**Objective** To quantify global relationships between sugar-sweetened beverage (SSB) intake and prices and examine the potential effectiveness of tax policy.

**Design** SSB intake data by country, age and sex from the Global Dietary Database were combined with gross domestic product and price data from the World Bank. Intake responsiveness to income and prices was estimated accounting for national income, age and sex differences.

**Setting** 164 countries.

**Population** Full adult population in each country.

**Main outcome measures** A consumer demand modelling framework was used to estimate the relationship between SSB intake and prices and derive own-price elasticities (measures of percentage changes in intake from a 1% price change) globally by age and sex. We simulated how a 20% tax would impact SSB intake globally. Tax policy outcomes were examined across countries by global income decile for representative age and sex subgroups.

**Results** Own-price responsiveness was highest in lowest income countries, ranging from −0.70 (p<0.100) for women, age 50, to −1.91 (p<0.001) for men, age 80. In the highest income countries, responsiveness was as high as −0.49 (p<0.001) (men, age 20), but was mostly insignificant for older adults. Overall, elasticities were strongest (more negative) at the youngest and oldest age groups, and mostly insignificant for middle-aged adults, particularly in middle-income and high-income countries. Sex differences were mostly negligible. Potential intake reductions from a 20% tax in lowest income countries ranged from 14.5% (95% CI: 29.5%, −0.4%) in women, 35 ≤ age < 60, to 24.9% (44.4%, 5.3%) in men, age ≥60. Intake reductions decreased with country income overall, and were mostly insignificant for middle-aged adults.

**Conclusions** These findings estimate the global price-responsiveness of SSB intake by age and sex, informing ongoing policy discussions on potential effects of taxes.

### Strengths and limitations of this study

► First study to examine sugar-sweetened beverage (SSB) intake and taxation in a global context, providing a better understanding of tax-policy effectiveness across the complete spectrum of countries.

► Results quantify the potential variability in influence of price on SSB intake across countries including by age and sex, suggesting that outcomes of SSB taxes may be significantly influenced by age and the income status of countries.

► Being a modelling study, the projected outcomes can only inform how taxes could affect behaviour.

► Cross-country analysis of this scope rely on specific data collection initiatives that often do not occur on an annual basis and/or do not provide specific variables; proxy variables are needed when data are not available.

other non-communicable diseases (NCDs).[1–5] Arguably, taxation is not punitive but market normalising, as the true costs of SSBs due to public healthcare expenditures and other societal costs from excessive intake are not reflected in current market prices. Thus, by increasing SSB prices relative to other foods, taxes can play a role in decreasing consumption, lowering societal costs and improving societal well-being.[6 7] Based on these considerations, a rapidly growing number of countries have implemented or announced national SSB taxes,[8 9] including Norway in 1981 and Samoa in 1984; Australia, French Polynesia, Fiji and Nauru between 2000 and 2007 and Finland, Hungary, France, Chile, Mexico, Barbados, St. Helena and Dominica since 2011. In 2018, the Philippines, the UK, South Africa, the Ireland, Peru and Norway implemented SSB taxes. Colombia and Saudi Arabia have included such taxes in recent proposals, while Bermuda, India

## INTRODUCTION

Taxation of sugar-sweetened beverages (SSBs) has received growing attention, given their links to excessive weight gain and increased risk of obesity, type-2 diabetes and

and Indonesia are considering similar measures. In the USA, more than 30 jurisdictions have implemented or attempted to pass SSB taxes since 2016, including San Francisco and Seattle in 2018.[10 11] Despite their growing acceptance globally, the potential impact of SSB taxation on intake remains uncertain, particularly how it might vary across countries, and by age and sex within countries.

Most studies of SSB taxation have been limited to a small group of countries or focused on a specific country or jurisdiction where taxes have been implemented.[12–17] No study to date has examined SSB consumption and taxation in a global context. In addition, few studies have considered how SSB intake could vary depending on the price of substitute products.[18] Because expert organisations are advocating and governments are considering SSB taxation across the globe,[19] examining demand in a global context can provide a better understanding of potential tax-policy effectiveness across the complete spectrum of countries, from most to least developed.

To investigate this issue, we examined SSB intake across 164 countries and estimated how intake differences within and across countries are influenced by the price of SSBs and substitute caloric beverages (fruit juice and milk), as well as other factors such as national income, age and sex. Based on WHO recommendations,[19] we further simulated how SSB intake would respond to a 20% tax (price increase). Tax-policy outcomes were examined across countries by income decile for representative age and sex subgroups.

## METHODS

Using globally representative intake and pricing data, we implemented a consumer demand modelling framework to examine determinants of SSB intake within and across countries. The modelling framework accounted for age and sex differences and economic determinants such as own price, price of substitutes (fruit juice and milk) and real per capita income at the national level. We also considered the potential for unmeasured region-specific differences, such as taste or other preferences, by including regional binary variables. Model estimates were used to derive SSB own-price elasticities for detailed strata (age, sex and countries by income decile), and to assess the potential impact of taxes on intake. Accounting for these factors, we report price elasticities of SSB intake (measures of the percentage change in intake from a 1% change in price), which have been a primary means of estimating potential tax-policy effectiveness.[20] We also evaluated the variability in tax-policy effectiveness and examined outcomes for select age and sex subgroups and countries by income decile.

### Data and sources

Data on SSB intake were derived from the 2010 Global Dietary Database (GDD), a database of global food and nutrient intakes by age (20 to 80 in 5 year intervals) and sex for 187 countries. The SSB category in the GDD includes intake of all sugar-sweetened beverages, including any beverage with added sugar and ≥50 kcal per 8 oz, such as carbonated beverages, sodas, energy drinks, fruit drinks, etc, excluding 100% juices. GDD data collection, statistical methods, data validation and findings have been described in detail (also see http://www.globaldietarydatabase.org/).[21–25] In brief, GDD data were derived based on national and subnational dietary surveys, informed by additional information from United Nations Food and Agricultural Organisation food balance sheets data, individual-level surveys from cohort studies, household expenditure surveys when dietary surveys were not available, as well as other data sources such as the WHO Global Infobase and the WHO STEPwise approach to Surveillance (STEPS) data.[25]

For prices, we used global price indices from the 2011 International Comparison Program (ICP) of the World Bank (see online supplemental table 1).[26 27] The ICP is a worldwide statistical initiative that produces price and expenditure data on consumer goods, services and capital goods. The price indices used in this study are standardised to a common currency, the US dollar in this case. Our choice of price variables was limited by inadequate data on a global scale. For instance, the ICP categories included milk but not SSBs and fruit juice. For SSBs, we used the ICP price index for sugar, which is justified, in part, due to sugar being a defining input. Similarly, we used the ICP fresh or chilled fruit price index as a proxy for fruit juice prices. Since sugar or fresh fruit may not be a major share of the final product price, particularly in rich countries, there are limitations to these proxies. In view of this, we adjusted the sugar and fresh fruit price indexes according to national income level using information on the value-added share of farm products in US food and beverage production (https://www.ers.usda.gov/data-products/food-dollar-series.aspx). This procedure resulted in relatively higher prices at higher income levels. Details are in the supplement (see online supplementary information, technical appendix).

We divided each price series by an aggregate price level index for *food and non-alcoholic beverages* to adjust for differences in overall food prices across countries. This discounts any price differences across countries due to differences in overall food costs and implicitly accounts for the cross-price effects of food products not in the model. The current analysis included 164 countries (4264 stratum observations) having both GDD intake and ICP price data.

For national income, we used 2010 gross domestic product (GDP) data expressed in US dollars per capita from the World Bank Development Indicators Database.[28] To account for differences in currency and purchasing power across economies, we used purchasing power parity (PPP) adjusted GDP. Since PPP-adjusted GDP accounts for inflationary factors across countries, we refer to our income measure as *real* per capita GDP. Income deciles were based on real per capita GDP for the 164 countries in the study.

## Model and analysis

To estimate SSB intake demand, we applied a single-equation framework and used a semi-logarithmic functional form (see online supplementary information, technical appendix).[29] [30] Many studies have used a double-log quadratic form.[31] However, a problem with the double-log form is that significant intake differences across subgroups can be lost in log conversions. A semi-log relationship allowed for a better assessment of subgroup effects on intake responsiveness. It has also been shown that semi-log models of demand are consistent with economic theory and contain the necessary information for obtaining, for instance, reliable measures of consumer welfare and the underlying preference structure of consumers.[29] Prior studies have also used a demand-system approach (multi-equation framework), primarily due to the need to account for the adding-up property when using expenditure data (ie, expenditures on all consumption categories 'add up' to total expenditures), which results in the error terms being correlated across categories. Since we are not estimating demand using an expenditure or allocation framework, the adopted approach is acceptable.

We accounted for age, sex and regional differences by allowing these factors to have a direct effect on intake, as well as an additional effect through income and prices, including a quadratic age term to allow for non-linear effects and the possibility of optimal responsiveness being between the youngest and oldest subgroups.

We accounted for varying preferences across countries due to factors not related to income or prices by including regional binary variables in the model: Southeast Asia, East Asia and High Income Asia Pacific (Asia) (13 countries); Central Europe, Eastern Europe and Central Asia (27 countries); Latin America and the Caribbean (LAC) (30 countries); Middle East, North Africa and South Asia (23 countries); sub-Saharan Africa (SSA) (45 countries) and High Income/Rest of World (HIC) (26 countries). HIC was comprised largely of western, industrialised countries; while not geographically connected, these countries share other similarities. We included several small island countries in this grouping because they were not sufficiently numerous to merit their own regional grouping (see online supplemental table 2).

We utilised F-tests to compare a model including all explanatory variables and interaction terms to a series of restricted models and arrived at the final parsimonious model. Least-squares regression treats data independently and does not account for within-country correlations resulting in biased and comparatively small standard errors. Correcting for this, all models were estimated assuming country clusters, that is, independent errors across countries but correlated errors within countries, as well as heteroskedastic-consistent errors.[32] The elasticities reported in the following section were derived using the estimated coefficients from model 3 (final model) (see online supplemental table 3).

Given WHO recommendations, we simulated how SSB intake would respond to a 20% tax (price increase).[19]

Results were evaluated across countries by income decile for the following demographic subgroups: men and women, age <35, 35 to 59, ≥60 years. We used probabilistic sensitivity analyses (Monte Carlo simulations) to derive 95% CIs of intake responsiveness to the tax. CIs were based on the covariance matrix of the estimated coefficients, which accounted for the variability in the own-price relationship and the additional variability due to age, sex and national income level.

## Patient and public involvement

Patients and the public were not involved in the design or planning of the study.

## RESULTS

### Global SSB intake

SSB intake levels varied significantly across countries (see online supplemental figure 1) and by world region and age (figure 1). LAC had the highest median intake at 311 g/day (men) and 288 g/day (women) – almost four times the intake in SSA, and six times the lowest intake region (Asia). Across age/sex strata globally, the group with the highest median intake was young men, age 20 (209 g/day), followed closely by young women, age 20 (188 g/day). Compared with 20 year olds, median global intake in men and women, age 80, was about 75% lower. Across age and sex strata worldwide, the highest intake level was observed for men, age 20, in Trinidad and Tobago (1239 g/day), and the lowest intake for women, age 80, in China (6 g/day). A more detailed discussion of global SSB intake by age, sex and world region is available.[33]

### SSB own-price elasticities

Given the variables in the final model, it was more appropriate to derive elasticities across country groups based on income level. We derived and compared SSB own-price elasticities across all strata jointly by age, sex and global income decile (figure 2 and table 1; also see online supplemental table 4). Note that reported values are derived at median intake levels by age and sex subgroup. Thus, observed differences across age, sex and income decile are solely a function of own-price interactions with sex, age and income. At any given age, SSB intake became less responsive to price changes with rising income. For instance, in women, age 20, the own-price elasticities ranged from −0.90 (p<0.001) for the lowest income decile to −0.47 (p<0.001) for the highest income decile. The decline in responsiveness became more pronounced with age. For instance, in men, age 80, the own-price elasticities ranged from −1.91 (p<0.001) for the lowest income decile to −0.43 (p>0.100) for the highest income decile. The influence of age on SSB own-price elasticities varied depending on income status. At lower income levels, elasticities were strongest (became more negative) at older ages; but at middle and higher income levels, there was less influence of age on elasticities. The least responsive

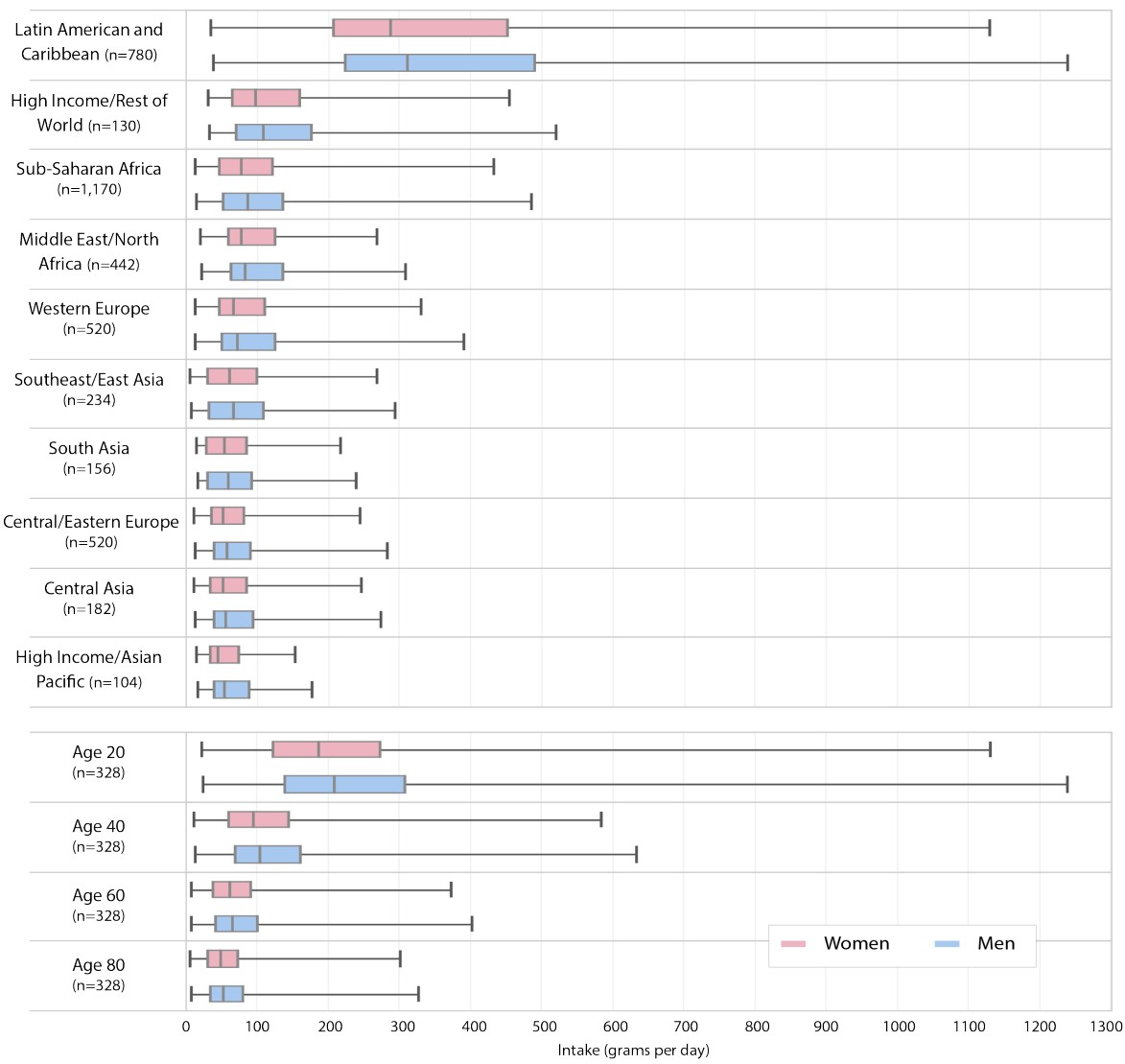

**Figure 1** Comparison of mean sugar-sweetened beverage intakes among adults in age, sex and country-specific strata across world regions and globally by select age groups. *n* represents the number of age, sex and country-specific subgroups in each stratum. Boxes represent the median intake value and IQR; error bars represent the minimum and maximum values. Source: Global Dietary Database, 2010.

group were middle-aged adults, particularly in upper-middle and higher income deciles.

### Potential impact of SSB taxes on intake

Potential reductions in median intake from a 20% tax (price increase) were largest for the lowest income decile, ranging from 14.5% (95% CI: −0.4 to 29.5) to 24.1% (5.3 to 44.4), depending on age and sex (table 2). Across income deciles, reductions varied less in younger adults (age <35) – for example, ranging from 16.8% (8.6 to 25.0) in young men in the lowest income decile to 7.9% (2.2 to 13.6) in the highest income decile – than in older adults (eg, men, age ≥60). This is consistent with the much higher baseline SSB intakes among younger adults globally (figure 1), suggesting that such intake will be significantly influenced by taxes regardless of income status. Older men and women (age ≥60) in the lowest income decile were estimated to be most influenced by SSB taxes, suggesting a high price-responsiveness to such

a luxury in poor nations globally. Insignificant outcomes were mostly observed for middle-aged and older adults in middle and higher income deciles.

### DISCUSSION

In this global analysis of SSB intakes and prices, we identified significant price responsiveness in nearly every age, sex and country income subgroup worldwide. We also identified significant heterogeneity in these potential responses. Price responsiveness was higher in lower income than in wealthier countries, consistent with expectations and the much higher relative share of income spent on food and other necessities in low-income countries. Interestingly, the response by age varied by national income. In lower income countries, own-price responsiveness increased with age, but less so in middle and higher income countries.

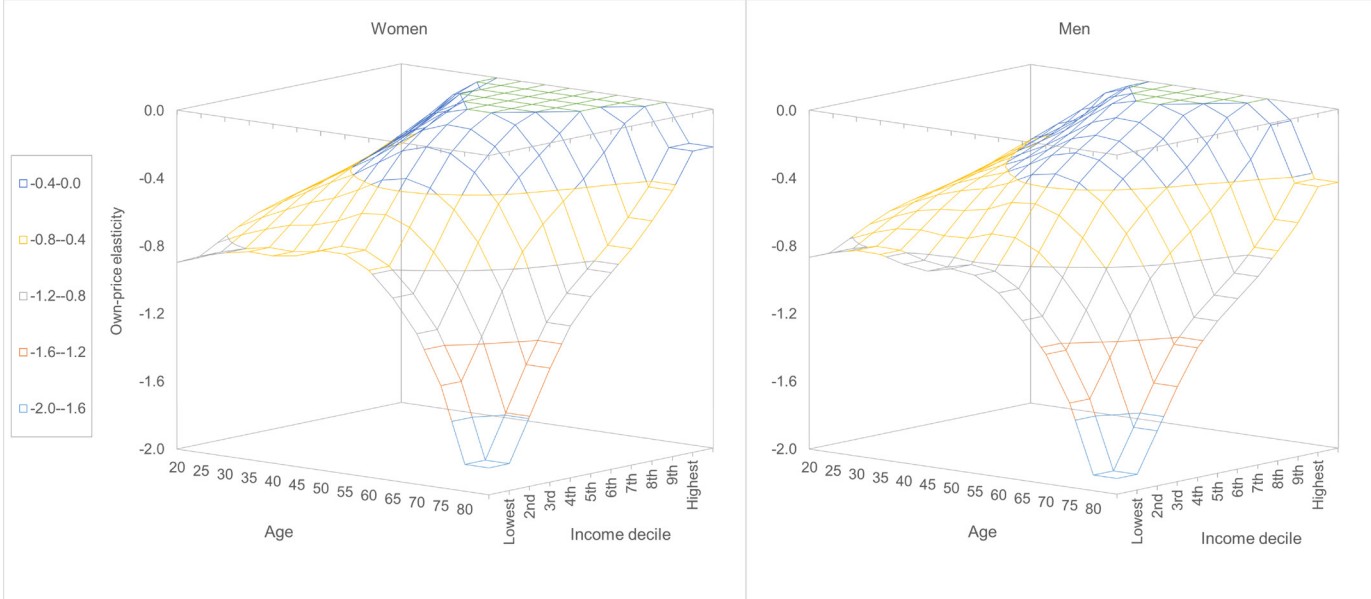

**Figure 2** Global sugar-sweetened beverage own-price elasticities by age, sex and global income decile. Values are derived at median intake levels by demographic subgroup. Own-price elasticities are based on 1% price changes. Income deciles are based on the national income of the 164 countries included in the study. Each decile is comprised of 16 countries (except the four lowest deciles, which are each comprised of 17 countries). The per capita income range (purchasing power parity-adjusted in thousand US$) for each decile: (1st) $0.6-$1.5, (2nd) $1.5-$2.7, (3rd) $2.7-$5.3, (4th) $5.5-$8.0, (5th) $8.3-$10.8, (6th) $11.1-$15.2, (7th) $15.3-$20.3, (8th) $20.6-$29.4, (9th) $30.4-$40.9 and (10th) $41.3-$127.2.

Finally, our estimates of effects of a 20% tax suggested significant SSB intake reductions across income levels, particularly for young adults. Outcomes for middle-aged adults, and older adults at higher income levels, were not significant.

### Strengths and limitations

This study has several strengths, the first being the extensive country coverage. We provide a global snapshot of SSB intake behaviour allowing for comparisons within and across most countries. Since past studies have been limited to a single country or a select group of countries, the results of this study inform policy and decision-making beyond the current state of knowledge. Problems associated with poor diets and NCDs occur in both developing and developed countries.[34] A comparative analysis across the complete spectrum of countries can assist international organisations in developing heterogeneous strategies for specific subgroups and countries. Our use of individual intakes by age, sex and country provides for more accurate representation of dietary behaviour. Previous findings based on expenditure data may be limited by differences in expenditures and actual consumption.

Potential limitations should also be considered. First, being a modelling study, the projected outcomes can only inform how taxes could affect behaviour. While an intervention study would be more fitting, interventions across 164 countries would not be feasible. Second, our analysis was limited by the use of price and income data at the national level. Ideally, our explanatory variables would also be at the subgroup level, reflecting that incomes

typically vary with age and sex, and different subgroups could face a different set of prices within a country. For instance, in countries where urban populations are relatively young, young adults could face different prices depending on market conditions in urban and rural areas. This limitation is due to the number of countries in our study. Such detailed data is not available for many countries.

While it would be ideal to have a time series of global SSB intake data, unfortunately these data do not exist. However, there is value in examining data at a point-in-time and intake in one demographic group compared with other groups, as well as comparing intake patterns across countries. Our purpose is to inform how demographic subgroups across countries might respond to price signals in form of taxes. There is value in understanding the relative responsiveness which can be gleaned from a cross-country snapshot.

The use of the global sugar prices as a proxy for SSB prices raises questions about the primary relationship of interest (SSB own-price elasticity). For higher income countries where farm production costs are a small share of the final product price, the proxy is less suitable and could result is lower 'own-price' responsiveness. Accordingly, we adjusted the price index to account for higher SSB prices relative to sugar prices at higher income levels. The adjustment resulted in a 10-fold to 15-fold increase in the index value for higher income countries similar to the USA. For low-income countries, adjusted and unadjusted prices were not that dissimilar (see online supplemental figure 2). Using adjusted prices, we found significantly

**Table 1** Own-price elasticities of SSB intake by age, sex and global income decile†

| Income decile‡ | Age 20 | Age 30 | Age 40 | Age 50 | Age 60 | Age 70 | Age 80 | Population-weighted average |
|---|---|---|---|---|---|---|---|---|
| | Women | | | | | | | |
| Lowest 10% | −0.90 (0.21)*** | −0.80 (0.25)*** | −0.78 (0.35)** | −0.70 (0.42)* | −0.78 (0.49) | −1.11 (0.53)** | −1.84 (0.60)*** | −0.82 (0.30)*** |
| 2nd | −0.83 (0.18)*** | −0.71 (0.21)*** | −0.65 (0.29)** | −0.54 (0.34) | −0.58 (0.38) | −0.88 (0.41)** | −1.59 (0.46)*** | −0.71 (0.25)*** |
| 3rd | −0.76 (0.16)*** | −0.62 (0.18)*** | −0.51 (0.23)** | −0.36 (0.27) | −0.37 (0.29) | −0.65 (0.30)** | −1.33 (0.34)*** | −0.59 (0.21)*** |
| 4th | −0.70 (0.14)*** | −0.54 (0.16)*** | −0.40 (0.20)** | −0.22 (0.22) | −0.19 (0.22) | −0.45 (0.21)** | −1.10 (0.24)*** | −0.49 (0.17)*** |
| 5th | −0.67 (0.14)*** | −0.49 (0.15)*** | −0.32 (0.18)* | −0.12 (0.19) | −0.07 (0.19) | −0.32 (0.17)* | −0.96 (0.18)*** | −0.40 (0.16)** |
| 6th | −0.64 (0.13)*** | −0.45 (0.14)*** | −0.26 (0.17) | −0.04 (0.18) | 0.02 (0.18) | −0.21 (0.15) | −0.84 (0.16)*** | −0.33 (0.15)** |
| 7th | −0.60 (0.13)*** | −0.41 (0.14)*** | −0.20 (0.17) | 0.04 (0.19) | 0.11 (0.18) | −0.11 (0.15) | −0.72 (0.16)*** | −0.27 (0.11)** |
| 8th | −0.57 (0.13)*** | −0.36 (0.14)** | −0.13 (0.18) | 0.13 (0.20) | 0.23 (0.20) | 0.02 (0.18) | −0.58 (0.19)*** | −0.22 (0.11)** |
| 9th | −0.53 (0.14)*** | −0.31 (0.15)** | −0.05 (0.20) | 0.23 (0.22) | 0.35 (0.24) | 0.16 (0.23) | −0.43 (0.24)* | −0.15 (0.06)** |
| Highest 10% | −0.47 (0.15)*** | −0.23 (0.17) | 0.06 (0.23) | 0.37 (0.27) | 0.52 (0.30) | 0.35 (0.31) | −0.22 (0.34) | −0.11 (0.07) |
| | Men | | | | | | | |
| Lowest 10% | −0.87 (0.19)*** | −0.79 (0.23)*** | −0.83 (0.32)*** | −0.81 (0.39)** | −0.91 (0.45)** | −1.24 (0.50)** | −1.91 (0.55)*** | −0.84 (0.30)*** |
| 2nd | −0.81 (0.17)*** | −0.71 (0.19)*** | −0.71 (0.27)*** | −0.66 (0.32)** | −0.73 (0.36)** | −1.03 (0.39)*** | −1.68 (0.43)*** | −0.76 (0.23)*** |
| 3rd | −0.75 (0.15)*** | −0.63 (0.16)*** | −0.59 (0.22)*** | −0.50 (0.25)** | −0.53 (0.27)* | −0.81 (0.28)*** | −1.44 (0.32)*** | −0.59 (0.21)*** |
| 4th | −0.69 (0.13)*** | −0.56 (0.14)*** | −0.48 (0.18)*** | −0.36 (0.20)* | −0.36 (0.21)* | −0.62 (0.20)*** | −1.24 (0.23)*** | −0.54 (0.16)*** |
| 5th | −0.66 (0.13)*** | −0.51 (0.13)*** | −0.41 (0.17)** | −0.27 (0.18) | −0.25 (0.18) | −0.50 (0.16)*** | −1.10 (0.19)*** | −0.40 (0.16)** |
| 6th | −0.63 (0.12)*** | −0.48 (0.13)*** | −0.35 (0.16)** | −0.19 (0.17) | −0.17 (0.17) | −0.40 (0.14)*** | −0.99 (0.17)*** | −0.42 (0.14)*** |
| 7th | −0.60 (0.12)*** | −0.44 (0.13)*** | −0.30 (0.16)* | −0.12 (0.17) | −0.08 (0.17) | −0.31 (0.14)** | −0.89 (0.16)*** | −0.28 (0.13)** |
| 8th | −0.57 (0.12)*** | −0.40 (0.13)*** | −0.23 (0.17) | −0.04 (0.18) | 0.02 (0.18) | −0.19 (0.17) | −0.76 (0.19)*** | −0.29 (0.12)** |
| 9th | −0.53 (0.13)*** | −0.35 (0.14)** | −0.16 (0.18) | 0.06 (0.21) | 0.14 (0.22) | −0.06 (0.21) | −0.62 (0.23)*** | −0.16 (0.09)* |
| Highest 10% | −0.49 (0.14)*** | −0.28 (0.15)* | −0.06 (0.21) | 0.19 (0.25) | 0.29 (0.28) | 0.11 (0.29) | −0.43 (0.31) | −0.16 (0.10) |

Values are derived at median intake levels by demographic subgroup. SEs are in (parenthesis). Population weights by sex, age and income status were obtained from the World Development Indicators Data Bank: https://databank.worldbank.org/data/reports.aspx?source=world-development-indicators#. *p≤0.10; **p≤0.05; ***p≤0.01.
*Price elasticities are based on 1% price changes. For instance, given a 1% SSB price increase in the lowest income countries, intake by women, age 20 falls by 0.90%. ‡Income deciles are based on the national income of the 164 countries included in the study. Each decile is comprised of 16 countries (except the four lowest deciles, which are each comprised of 17 countries). The per capita income range (PPP-adjusted in thousand US$) for each decile: (1st) $0.6-$1.5, (2nd) $1.5-$2.7, (3rd) $2.7-$5.3, (4th) $5.5-$8.0, (5th) $8.3-$10.8, (6th) $11.1-$15.2, (7th) $15.3-$20.3, (8th) $20.6-$29.4, (9th) $30.4-$40.9 and (10th) $41.3-$127.2.
PPP, purchasing power parity; SSB, sugar-sweetened beverage.

higher own-price responsiveness compared with estimates using unadjusted prices.

### Comparison with other studies

Since previous research has mostly focused on higher income countries, primarily the USA, it is difficult to compare all of our results with earlier findings. Several US based studies have considered how SSB consumption would respond to a tax. Given a 10% tax, the projected decrease in SSB sales ranged from 6.7% to 18.2%.[15] These results are greater than our findings for middle-aged and older adults in the highest income decile, but are closer to our findings for young adults (7.3%, women, age <35, and 7.9%, men, age <35), although we are considering a 20% tax.

Our tax outcomes are due to comparably smaller own-price elasticities. Whereas our own-price elasticity estimates for the highest income countries range from −0.5 to −0.0, meta-analyses of US studies give estimates of −0.8 (-3.2 to −0.13) and −1.1 (-1.3 to −0.9).[16 35] In a

study of Mexico using data before and after implementation of a national soda tax (10%) in 2014, SSB purchases decreased by an average of 6% during the first year of implementation,[12] which is actually comparable to our findings for young adults in middle-income countries. Other studies of Latin American countries using household survey data reported estimates more comparable to our results for lower income countries.[36–38]

The fact that our estimates are relatively smaller does not necessarily make them less accurate. Note that past studies have mostly used expenditure data. It has been documented that significant changes in expenditures do not always result in changes in the quantity or quality of food consumption.[39] In fact, studies have found the association between food expenditures and intake to be particularly weak and insufficient for diet and nutrition research.[40] For instance, a recent study of the SSB tax in Berkeley, California, USA, found significant reductions in consumer spending on SSBs, increased spending on

**Table 2** Potential impact of a 20% tax (price increase) on SSB intake by age, sex and global income decile.

| Income decile* | Women age <35 | Men age <35 | Women 35 ≤ age < 60 | Men 35 ≤ age < 60 | Women age ≥60 | Men age ≥60 |
|---|---|---|---|---|---|---|
| | Percentage change in intake (95% CI) | | | | | |
| Lowest 10% | −17.1 (−26.1 to −8.1) | −16.8 (−25.0 to −8.6) | −14.5 (−29.5 to 0.4) | −15.9 (−29.5 to −2.2) | −22.3 (−43.2 to −1.4) | −24.9 (−44.4 to −5.3) |
| 2nd | −15.6 (−23.3 to −7.9) | −15.4 (−22.5 to −8.3) | −11.6 (−23.8 to 0.7) | −13.2 (−24.3 to −1.9) | −17.7 (−33.9 to −1.4) | −20.6 (−35.9 to −5.3) |
| 3rd | −14.0 (−20.6 to −7.4) | −14.0 (−20.1 to −7.8) | −8.5 (−18.3 to 1.3) | −10.4 (−19.4 to −1.4) | −13.0 (−24.8 to −1.2) | −16.3 (−27.4 to −5.1) |
| 4th | −12.6 (−18.5 to −6.7) | −12.7 (−18.2 to −7.3) | −5.9 (−14.0 to 2.2) | −8.0 (−15.5 to −0.5) | −9.0 (−17.3 to −0.7) | −12.5 (−20.5 to −4.6) |
| 5th | −11.7 (−17.3 to −6.2) | −11.9 (−17.1 to −6.7) | −4.2 (−11.6 to 3.2) | −6.5 (−13.2 to 0.3) | −6.3 (−12.9 to 0.3) | −10.1 (−16.4 to −3.7) |
| 6th | −11.0 (−16.4 to −5.6) | −11.3 (−16.3 to −6.2) | −2.8 (−9.9 to 4.3) | −5.2 (−11.6 to 1.3) | −4.2 (−10 to 1.7) | −8.1 (−13.7 to −2.4) |
| 7th | −10.3 (−15.7 to −5.0) | −10.6 (−15.6 to −5.7) | −1.4 (−8.5 to 5.6) | −4.0 (−10.4 to 2.5) | −2.1 (−8 to 3.9) | −6.2 (−11.8 to −0.4) |
| 8th | −9.5 (−14.9 to −4.1) | −9.9 (−14.9 to −4.9) | 0.2 (−7.3 to 7.6) | −2.5 (−9.2 to 4.3) | 0.4 (−6.6 to 7.5) | −3.8 (−10.4 to 2.8) |
| 9th | −8.5 (−14.2 to −2.9) | −9.0 (−14.3 to −3.8) | 2.0 (−6.3 to 10.3) | −0.8 (−8.3 to 6.7) | 3.2 (−5.8 to 12.3) | −1.2 (−9.6 to 7.2) |
| Highest 10% | −7.3 (−13.5 to −1.1) | −7.9 (−13.6 to −2.2) | 4.5 (−5.4 to 14.3) | 1.4 (−7.6 to 10.3) | 7.0 (−5.3 to 19.3) | 2.2 (−9.1 to 13.6) |

Values are reductions from median intake levels for each demographic subgroup.
*Income deciles are based on the national income of the 164 countries included in the study. Each decile is comprised of 16 countries except the four lowest deciles, which are each comprised of 17 countries. The per capita income range (PPP-adjusted in thousand US$) for each decile: (1st) $0.6-$1.5, (2nd) $1.5-$2.7, (3rd) $2.7-$5.3, (4th) $5.5-$8.0, (5th) $8.3-$10.8, (6th) $11.1-$15.2, (7th) $15.3-$20.3, (8th) $20.6-$29.4, (9th) $30.4-$40.9 and (10th) $41.3-$127.2.
PPP, purchasing power parity; SSB, sugar-sweetened beverage.

substitute beverages, but insignificant reductions in reported SSB intake.[41] Another issue is that SSBs are less perishable than other foods. When goods have an extended shelf life, individuals can take advantage of price discounts, increasing expenditures when prices are low, stock piling for future consumption. Ignoring this fact can result in overestimates of own-price elasticities.[42]

## CONCLUSION

This is the first study to examine SSB consumption and taxation in a global context. Our findings provide a better understanding of the potential effectiveness of taxes across the full spectrum of countries. Overall, we found that the influence of SSB prices on intake significantly depends on the income status of countries. Our results suggest that intake reductions (in per cent) could be small or negligible for certain demographics in higher income countries. Although small in percentage terms, actual intake reductions could still be sizeable enough for high-consuming subgroups for taxes to be worth pursuing. For higher income countries, a larger tax or a tax combined with other approaches might be needed to significantly change behaviour. For instance, taxes could be combined with media and education campaigns, food labelling and other interventions.[43] For all adults in lower income countries and young adults globally, our findings indicate that taxes would be particularly effective, which is to be expected since food expenditures account for a greater share of income for these groups making them more sensitive to prices.

**Contributors** AM conceptualised the study and was responsible for the study design, model estimations and contributed to the interpretation of results. AM, DM and BM contributed to the interpretation of results and discussion. DM wrote and edited sections describing the intake data. BM was primarily responsible for the literature review and facilitated the data agreement with the International Comparison Program, World Bank. DM provided the intake data and obtained the funding. DRM and AM were responsible for the visualisations and corresponding text. AM was the primary author, but all authors contributed to writing the manuscript. AM is the manuscript's guarantor.

**Funding** The Bill & Melinda Gates Foundation (project: Global Dietary Habits among Women, Price and Income Elasticities and Validity of Food Balance Sheets).

**Disclaimer** The findings and conclusions in this publication are those of the author(s) and is not representative of official US Department of Agriculture or US Government determination or policy.

**Competing interests** None declared.

**Patient consent for publication** Not required.

**Provenance and peer review** Not commissioned; externally peer reviewed.

**Data sharing statement** No additional data available.

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
