## [Reviewer comments · BMJ Open]

ARTICLE DETAILS

TITLE (PROVISIONAL)	Global patterns in price elasticities of sugar-sweetened beverage intake and potential effectiveness of tax policy: a cross-sectional study of 164 countries by sex, age, and global-income decile
AUTHORS	Muhammad, Andrew; Meade, Birgit; Marquardt, David; Mozaffarian, Dariush

VERSION 1 – REVIEW

REVIEWER	Lennert Veerman Griffith University, Australia I would like to mention that I have reviewed this paper previously for another journal. The paper hasn't really changed, and therefore my comments are similar to those a year ago.
REVIEW RETURNED	30-Oct-2018

GENERAL COMMENTS	This paper addresses the topical issue of the potential impact of taxing sugar-sweetened beverages (SSBs) on consumption of those drinks, and substitute drinks juice and milk, for all countries of the world. The results indicate own-price elasticities that are considerably lower than other studies in this area, and this seems to be due to the choice of data and the structure of the model used to do the analyses. Cross-price elasticities are higher than other studies – though there are few of those and results vary widely. I have four major concerns about the analysis. First, in essence, in epidemiological terms, this is an ecological study. It tries to predict SSB consumption based on various data from 164 countries around the world at a single point in time. I don't think this is a very strong design to answer this type of question. Furthermore, the data were from a variety of different sources, with differing biases and errors. Second, the authors used the price of sugar as a substitute for the price of SSBs (and that of fresh fruit for fruit juice). I don't think that choice is warranted. As the authors note in the discussion, this may lead to underestimation of the magnitude of own-price elasticities. Sugar is only one of the inputs that determines the price of SSBs, but far from the only one. Other ingredients, but more importantly, the distribution chain, would be other factors of relevance. A doubling of the sugar price might add only 10% to the overall cost of the drink to the consumer. Therefore, even a perfect correlation between sugar price and sugary drinks prices does not make the unadjusted sugar price a good proxy for the price of sugared drinks; not without adjustment, and I did not read of any such adjustment. Since material costs would tend to make up a
---

	greater proportion of the price of drinks relative to labour costs, the own-price elasticity would be more biased towards the null in high-income countries, which is conform the results presented in this paper. Third, I don't quite understand how the regional binary variables work out in the model. Supplemental Table 2 shows that in high-consuming regions (e.g., LAC), these add a fixed quantity of consumption, and in regions with low levels of consumption (e.g., Asia), add it, all on the backdrop of a large constant that is positive in models 1 and 2 but negative in model 3. For a region like LAC, this seems to leave only a proportion of SSB consumption that can vary as a function of price. If that variation is the same (or similar) across all regions in an absolute sense (grams per % increase in price), this translates to a low PE in regions with a high SSB consumption (e.g. LAC) and a high one in regions with a low consumption (e.g. Asia). Figure 2 does indeed show that pattern. Similarly, younger people (with greater consumption) would have lower estimated price elasticities than older persons, and indeed that is what we see in the results. I am concerned this may be an artefact – and so are the authors, given their comments on page 15 where they seem to suggest ignoring some of their own results (“Consequently, the small own-price elasticities for LAC may not be representative of individual behavior in the region”). Finally, I have concerns about the cross-price elasticities. Where the own-PEs are lower than most other studies found, the cross-PEs are higher, especially for juice. I can't quite fathom the reason for this, but it may have to do with the fact that own price elasticities are underestimated because they are based on the price of sugar, not SSBs.
--	---

REVIEWER	Carlos Manuel Guerrero-López National Institute of Public Health, Mexico
REVIEW RETURNED	20-Nov-2018

GENERAL COMMENTS	This papers addresses an important research question. However, there are several issues that deserve attention. Here I list my main observations: Abstract. I suggest to include a brief description of the data used in the study. Introduction. I suggest to update the review of taxes implemented accross several countries. For instance, use this: https://www.cambridge.org/core/journals/public-health-nutrition/article/sugarsweetened-beverage-taxes-in-2018-a-year-of-reflections-and-consolidation/6D57123CE77918D3A87051CE38DA5C13 Methods. Consumption data seem to be from 2010 and prices from 2011. Is that ok? If so, how does that change the interpretation of the results? If not, I suggest to make it clear that data on consumption and prices are from the same year. Results. Tables 1 and 2: I suggest to include a "total" row and column to inform the price elasticities at an aggregated level.
--

	Figure 2. I suggest to inform the mean own-price elasticity on the figure. For instance, you mention that figure regarding Central Asia. Including the figures is important in order to report the results. Discussion. In the last paragraph of page 15 (comparison with other studies), you mention a study from Mexico about a reduction of 6% in consumption after an implementation of a national soda tax of 10 %, "which is closer, but still larger than our findings for middle-income countries". However, literature about price-elasticities of SSB in Mexico, Chile and Ecuador show that the demand for SSB is elastic (price elasticities greater than 1 in absolute value). These estimates use specific ad-hoc country-level data that may be better than the data that you used. See for example these papers: https://journals.plos.org/plosone/article?id=10.1371/journal.pone.0152260 https://www.ncbi.nlm.nih.gov/pubmed/26386463 https://bmcpublichealth.biomedcentral.com/articles/10.1186/s12889-017-4098-x How can you discuss the difference with your results beyond the nature of SSB as storable goods?
--	---

REVIEWER	Cherry Law London School of Hygiene and Tropical Medicine, United Kingdom
REVIEW RETURNED	26-Nov-2018

GENERAL COMMENTS	This paper studied the global patterns of sugar-sweetened beverage (SSB) intake using Global Dietary Database (GDD). Together with price and national income data, it estimated demand elasticities and used these estimates to analyse the potential impact of a 10% SSB tax. Considering the rising popularity of SSB taxes, how the effectiveness of such a tax may differ across countries is an important research question. However, there are some issues with the technical specification in this paper. Comments:  - The authors need to strengthen the justifications for the use of semi-logarithm equation. They argue that since they are not using expenditure data and the correlation across categories does not exist with individual intakes, it is acceptable to not use demand system approach. However, as stated on Page 7, some data of GDD is derived from expenditure data when dietary surveys are not available. More importantly, the error correlation across food groups continue to exist even with individual intake data as the intake level of milk / fruit juice is likely to affect the SSB intake level. Is there any reference to support the argument that "the adapted approach is acceptable" (line 45-49 on page 8)? Also, the authors could consider using log-log functional form which is a more commonly used alternative to demand system than semi-log form. - Issues related to the equations:  1. "age" is not a continuous variable as it is of 5-year interval. It should be used as categorical variable. The quadratic term of age is thus problematic. 2. If the linear income term is interacted with price variables, shouldn't the quadratic income term be interacted with price too?
---

	3. It is unclear how equation 2 is related to equation 1. My understanding is that for each coefficient in equation 1 is further expanded as equation 2. However, this doesn't seem to be the case reflected in the final results as the age variable is interacted with the region variable, which is not discussed in the model. 4. Why is that cross price elasticity does not depend on income but own-price elasticity does? 5. Based on equation 5, expenditure share data is needed to calculate cross price elasticity. However, there is no discussion on the source of this data. 6. Is there any evidence to support the argument that "share of SSB expenditure in total income is likely to be small?" 7. The use of sugar and fruit prices as proxies implicitly assumes that cross-country price differences in these products are strongly correlated to that in SSB and fruit juices. The authors justified it using sugar and SSB prices in the US. However, this only provided evidence for a time series correlation but does not give support to the cross-country correlation between sugar and SSB prices. - This paper is a country level analysis based on aggregated data which has limited explanatory power on individual behaviour. The argument in line 26-29 in the conclusion saying "from which we could also infer the behaviour of lower income individual" is thus not justified.
--	---

VERSION 1 – AUTHOR RESPONSE

Reviewer: 1

Reviewer Name: Lennert Veerman

Institution and Country: Griffith University, Australia

'None declared'

General reply:

We thank the reviewer for his comments and concerns and have made all efforts to incorporate suggestions and address concerns in the revised draft.

The benefit of this study is the country coverage and identification of international consumption patterns and determinants. A drawback of any global or cross-country analysis is that researchers are limited to specific data collection initiatives that often do not occur on an annual basis and/or do not provide detailed or specific variables. Needless to say, data used for cross-country analysis should not be held to the same standard as a single-country or household-level study where detailed data are readily available. For instance, if country-specific soda, bottled water, or fruit juice price indexes were readily available, we would have included them in the analysis. As these prices do not exist on a

global scale, we believe the cost of using second best prices to be lower than losing the benefit of the insight this global study brings to international policy makers. That said, your critique is still warranted and we address this issue in the current draft.

As suggested, we modified the sugar and fresh fruit price indexes to reflect production cost differences across countries. The use of “adjusted” prices resulted in higher estimates but a similar pattern as our previous findings. See the supplemental technical appendix for details.

While the concerns raised by the reviewer are valid, they would be more applicable if our results showed little or no significance, or if our results were counter to economic intuition. However, we do find clear and robust evidence of significant relationships between global SSB intake and prices. Furthermore, our results are also consistent with economic intuition and consumer behavior as they relate to age, income status, and consumer preferences.

(Reviewer’s comments are *italics*)

1. *The results indicate own-price elasticities that are considerably lower than other studies in this area, and this seems to be due to the choice of data and the structure of the model used to do the analyses. Cross-price elasticities are higher than other studies – though there are few of those and results vary widely.*

The reviewer is correct. This a function of the choice of data, but the choice of model is not necessarily an issue in this context. Although the elasticities derived using this model inversely depend on intake levels, the results presented in this study assume median intake levels by subgroup. Thus, the observed differences in both the elasticities and tax simulations are due more so to the sex, age, and income interactions. As far as the data are concerned, our lower estimates are due, in part, to using actual intakes instead of expenditures, the latter are likely more responsive to prices. We do acknowledge, however, that this could also be due to the use of sugar prices as a proxy and have adjusted prices accordingly. While our estimates are still lower by comparison, they are now more in line with the literature.

Since the cross-price elasticities are not important to the overall analysis and are not comparable due to the added own-price/income interaction term, we have decided not to report them in the paper to avoid confusion.

2. *In essence, in epidemiological terms, this is an ecological study. It tries to predict SSB consumption based on various data from 164 countries around the world at a single point in time. I don’t think this is a very strong design to answer this type of question.*

We acknowledge this as a study limitation and included the following in the limitations section of the paper.

...While it would be ideal to have a time series of global SSB intake data, unfortunately these data do not exist. However, there is value in examining data at a point-in-time and intake of one demographic group within a country compared to the other groups, as well as comparing intake across countries. The purpose of this study is to get a sense of how countries, regions, demographic groups might respond more or less to price signals. There is value in understanding the relative responsiveness that can be gleaned from a cross-country snapshot.

3. Furthermore, the data were from a variety of different sources, with differing biases and errors.

It is not clear why this is an issue since numerous econometric studies are based on data from different sources. As mentioned in the general reply, this would be more of a concern if results were mostly insignificant or counter to economic intuition or rational behavior.

4. The authors used the price of sugar as a substitute for the price of SSBs (and that of fresh fruit for fruit juice). I don't think that choice is warranted. As the authors note in the discussion, this may lead to underestimation of the magnitude of own-price elasticities. Sugar is only one of the inputs that determines the price of SSBs, but far from the only one. Other ingredients, but more importantly, the distribution chain, would be other factors of relevance. A doubling of the sugar price might add only 10% to the overall cost of the drink to the consumer. Therefore, even a perfect correlation between sugar price and sugary drinks prices does not make the unadjusted sugar price a good proxy for the price of sugared drinks; not without adjustment, and I did not read of any such adjustment. Since material costs would tend to make up a greater proportion of the price of drinks relative to labor costs, the own-price elasticity would be more biased towards the null in high-income countries, which is conform the results presented in this paper.

See our general reply.

5. I don't quite understand how the regional binary variables work out in the model.

Supplemental

Table 2 shows that in high-consuming regions (e.g., LAC), these add a fixed quantity of consumption, and in regions with low levels of consumption (e.g., Asia), add it, all on the backdrop of a large constant that is positive in models 1 and 2 but negative in model 3. For a region like LAC, this seems to leave only a proportion of SSB consumption that can vary as a function of price. If that variation is the same (or similar) across all regions in an absolute sense

(grams per % increase in price), this translates to a low PE in regions with a high SSB consumption (e.g. LAC) and a high one in regions with a low consumption (e.g. Asia). Figure 2 does indeed show that pattern. Similarly, younger people (with greater consumption) would have lower estimated price elasticities than older persons, and indeed that is what we see in the results. I am concerned this may be an artifact – and so are the authors, given their comments on page 15 where they seem to suggest ignoring some of their own results (“Consequently, the small own-price elasticities for LAC may not be representative of individual behavior in the region”).

Note that in the final model, the regional dummy variables do not interact with any of the price terms. We arrived at this model because region interactions were highly insignificant in a more

expansive model; based on F-tests, we excluded these terms from the model. Consequently, the binary regional terms in the final model do not in any way affect the elasticity estimates. What is being observed is that the elasticity values inversely depend on the level of intake. All else equal, this should be the case intuitively (i.e., changes in intake are larger in percentage terms when intake is low).

To avoid confusion, we only report elasticities assuming median intake levels, which is a common practice when reporting elasticities.

With the following statement “*Consequently, the small own-price elasticities for LAC may not be representative of individual behavior in the region*” we are not ignoring our own results, but are simply suggesting that based on our final model, elasticities derived using median (or mean) intake levels by sex, age, and national income are more appropriate, since these are the only terms interacting with prices.

6. *Finally, I have concerns about the cross-price elasticities. Where the own-PEs are lower than most other studies found, the cross-PEs are higher, especially for juice. I can't quite fathom the reason for this, but it may have to do with the fact that own price elasticities are underestimated because they are based on the price of sugar, not SSBs.*

The relatively higher cross-price elasticities was primarily due to the positive estimate on the own-price/income interaction term resulting in relatively lower own-price elasticities with at higher income levels. Note that we did not interact income with fruit or milk prices.

Since the cross-price elasticities are not important to the overall analysis and are not comparable due to the added own-price/income interaction term, we have decided not to report them in the paper to avoid confusion.

Reviewer: 2

Reviewer Name: Carlos Manuel Guerrero-López

Institution and Country: National Institute of Public Health, Mexico

Please state any competing interests or state 'None declared': Non declared

1. Abstract. I suggest to include a brief description of the data used in the study.

Noted and updated.

2. Introduction. I suggest to update the review of taxes implemented across several countries. For instance, use this: <https://www.cambridge.org/core/journals/public-healthnutrition/article/sugarsweetened-beverage-taxes-in-2018-a-year-of-reflections-andconsolidation/6D57123CE77918D3A87051CE38DA5C13>

We appreciate the reference on the current state of SSB tax implementations. The reference and new information have been added to the paper.

3. Methods. Consumption data seem to be from 2010 and prices from 2011. Is that ok? If so, how does that change the interpretation of the results? If not, I suggest to make it clear that data on consumption and prices are from the same year.

Given how the GDD and ICP data are derived there are some overlapping years. For instance, the ICP 2011 price data for OECD countries were derived using data from 2009 and 2010 as well as 2011. That said, since we are only estimating associations and are not suggesting causality, this may not be an issue. Also, it is reasonable to assume that price differences across countries in 2011 are not that different from 2010. The other option would be to use 2005 ICP price data which includes fewer countries.

4. Results. Tables 1 and 2: I suggest to include a "total" row and column to inform the price elasticities at an aggregated level.

We now include a population weighted average in Table 1.

Table 2 are percentage declines; a total measure would require that we convert our percentage changes to gram equivalents, which are a function of how SSB intake data are reported in the GDD. Given that we are more focused on informing the behavior of different groups, we are somewhat hesitant to infer from our tax results global or total impacts.

5. Figure 2. I suggest to inform the mean own-price elasticity on the figure. For instance, you mention that figure regarding Central Asia. Including the figures is important in order to report the results.

Based on an issue raised by another reviewer and given that the own price elasticities do not depend on region (results indicated that once age, sex, and national income are accounted for, region interactions with prices were insignificant), we no longer report the regional elasticities.

6. Discussion. In the last paragraph of page 15 (comparison with other studies), you mention a study from Mexico about a reduction of 6% in consumption after an implementation of a national soda tax of 10 %, "which is closer, but still larger than our findings for middle-income countries". However, literature about price-elasticities of SSB in Mexico, Chile and Ecuador show that the demand for SSB is elastic (price elasticities greater than 1 in absolute value). These estimates use specific ad-hoc country-level data that may be better than the data that you used.

See for example these papers:

<https://journals.plos.org/plosone/article?id=10.1371/journal.pone.0152260>

<https://www.ncbi.nlm.nih.gov/pubmed/26386463>

<https://bmcpublichealth.biomedcentral.com/articles/10.1186/s12889-017-4098-x>

All 3 studies were included in the paper to indicate that higher estimates do exist in the literature.

7. How can you discuss the difference with your results beyond the nature of SSB as storable goods?

We have added more discussion on why estimates based on expenditures are larger and included additional references supporting this case. We also provide evidence specific an SSB tax study in Berkeley California.

Reviewer: 3

Reviewer Name: Cherry Law

Institution and Country: London School of Hygiene and Tropical Medicine, United Kingdom

Please state any competing interests or state 'None declared': None Declared

We thank the reviewer for her comments and have made all efforts to incorporate suggestions and address any concerns in the revised draft.

1. The authors need to strengthen the justifications for the use of semi-logarithm equation. They argue that since they are not using expenditure data and the correlation across categories does not exist with individual intakes, it is acceptable to not use demand system approach. However, as stated on Page 7, some data of GDD is derived from expenditure data when dietary surveys are not available. More importantly, the error correlation across food groups continue to exist even with individual intake data as the intake level of milk / fruit juice is likely to affect the SSB intake level. Is there any reference to support the argument that "the adapted approach is acceptable" (line 45-49 on page 8)?

Also, the authors could consider using log-log functional form which is a more commonly used alternative to demand system than semi-log form.

We have added language and references to strengthen the justification for using the semi-log functional form.

The following has been added to the paper (with references)...

"...Many studies have used a double-log quadratic form. However, a problem with the double-log form is that significant intake differences across subgroups can be lost in log conversions. A semi-log relationship allowed for a better assessment of subgroup effects on intake responsiveness. It has also been shown that semi-log models of demand are consistent with economic theory and contain the necessary information for obtaining, for instance, reliable measures of consumer welfare and the underlying preference structure of consumers..."

The cross-equation error issue is primarily due to the fact that analysis is often based on expenditure allocations. Thus, the n th relationship could be derived from $n-1$ equations. If an explanatory variable in our model was total beverage intake, or if our dependent variable was SSB intake share of total beverage (or food) intake, then cross-equation correlated errors would be an issue because a change in our dependent variable implies a change in a related variable. In our model, intake is being explained by total income and prices.

Since we are not estimating demand using an intake allocation or share equation framework, a system approach is not needed. That said, as long as there are no cross-equation parameter restrictions and regressors are potentially the same across equations, least squares on a single equation is equivalent to SUR or MLE on a system. So, even in the context of an equation system, least squares on an individual equation yield identical results.

2. "age" is not a continuous variable as it is of 5-year interval. It should be used as categorical variable. The quadratic term of age is thus problematic.

This has been noted and the language changed.

This is a reasonable but touchy point because *age* is still a numerical variable and 5-year intervals are still relatively small. Your point is clearly valid for most categorical variables, which are qualitative and assigned numbers usually do not mean anything in terms of value. We realize that our age variable could be restated as binary variables (e.g., Age20=1 if 19<age<24; 0 otherwise). Unfortunately, the same could also be said of age in one-year intervals (e.g., Age20=1 if 19<age<21; 0 otherwise).

Many of our age-squared terms were highly significant, age-squared was key in defining the curvature of price elasticity estimates, and the results are easily derived and explained with actual age values. This suggests that our approach was not too problematic in that our results are not nonsensical.

3. If the linear income term is interacted with price variables, shouldn't the quadratic income term be interacted with price too?

This interaction was in the unrestricted model but was highly insignificant and F-tests indicated that we delete it from the model. As stated in the previous comment, we were limited by available degrees of freedom. When accounting for within country correlations, you are left with only 164 "unique" observations. Thus, within reason, we excluded any unnecessary terms.

4. It is unclear how equation 2 is related to equation 1. My understanding is that for each coefficient in equation 1 is further expanded as equation 2. However, this doesn't seem to be the case reflected in the final results as the age variable is interacted with the region variable, which is not discussed in the model.

We have update the text to reflect this point.

5. Why is it that cross price elasticity does not depend on income but own-price elasticity does?

See reply to comment 3.

6. Based on equation 5, expenditure share data is needed to calculate cross price elasticity. However, there is no discussion on the source of this data. Is there any evidence to support the argument that "share of SSB expenditure in total income is likely to be small?"

Since the cross-price elasticities are not important to the results and are not comparable due to the added own-price/income interaction term, based on a reviewer, we have decided to not report them to avoid confusion.

7. The use of sugar and fruit prices as proxies implicitly assumes that cross-country price differences in these products are strongly correlated to that in SSB and fruit juices. The authors justified it using sugar and SSB prices in the US. However, this only provided evidence for a time series correlation but does not give support to the cross-country correlation between sugar and SSB prices.

As suggested by a reviewer, we modified the sugar and fresh fruit price indexes to reflect production cost differences across countries and that final product prices are significant higher than primary input prices in higher income countries. The use of adjusted prices results in larger estimates but a similar pattern as our previous findings. See the supplemental technical appendix for details.

Consequently, we no longer give justification based the U.S. time series.

8. This paper is a country level analysis based on aggregated data which has limited explanatory power on individual behaviour. The argument in line 26-29 in the conclusion saying "from which we could also infer the behaviour of lower income individual" is thus not justified.

Noted and deleted.

VERSION 2 – REVIEW

REVIEWER	Lennert Veerman Griffith University, Australia
REVIEW RETURNED	29-Jan-2019

GENERAL COMMENTS	The authors have clearly done their best to improve the paper on the basis of reviewer comments. They have improved the analysis by adjusting for the fact that sugar price is only one determinant of the overall consumer price of SSBs – though I do wonder if doing so using information on the value-added share of farm products in U.S. food and beverage production is appropriate for these global data. I also think the decision to sacrifice the estimates of cross-price elasticities was a good one. But even with the above adjustment, the price elasticity estimates are still far lower than almost any other study. (The highest results for US age groups are about half the size of the lowest findings from several other US studies, as the discussion makes clear.) While it is certainly true that those other studies also have their flaws, I still think a study based on data from around the world in a single year that uses adjusted proxy data for its independent variable is unlikely to provide better estimates. And that is for the US, where the adjustment factor was from. My interpretation is that the study shows that after adjustments, sugar prices are correlated with the intake of SSBs, and sheds some light on which age groups and countries are likely to be more responsive to changes in the (relative) price of sugar, but says little about the absolute magnitude of those responses. The questions, then, are whether this study adds enough to the literature to warrant publication, and whether the findings have been adequately put into context by the discussion. I am not sure about the first point. On the second point, the authors do acknowledge weaknesses, and show (as per above) that their estimates of own-price elasticities are much below those of other studies. The discussion does not mention any of these caveats, however. Whether these PEs are sufficiently strong to take the next step and estimate the likely impact of SSB taxes is moot. I don't think so; I would keep it at observing that the young and those in low-income countries are likely to be most responsive to such taxes. PS: While the Estonian government proposed a tax on SSBs that would have come into force by now, and Parliament voted in favour, the country still does not have an SSB tax. In a very unusual move, the President refused to sign, questioning the constitutionality of the exemption for SSBs sold on the ferries to and from the country. The tax was later deemed not to violate the constitution, but the timetable no longer tenable. With that, the matter rested with the Parliament again, but with elections in sight the government parties chose not to pick another fight with a powerful vested interest. So there the matter rests.
---

REVIEWER	Carlos Manuel Guerrero-López National Institute of Public Health, Mexico
REVIEW RETURNED	04-Feb-2019

GENERAL COMMENTS	Thank you for your replies. I agree with the answers the authors gave.
--

REVIEWER	Cherry Law London School of Hygiene and Tropical Medicine
-----------------	--

REVIEW RETURNED	17-Jan-2019
-------------

GENERAL COMMENTS	While the authors have greatly improved the paper, there are some issues remained: 1. I appreciated the efforts the authors made to adjust the prices of sugar and fruits for production cost difference across countries. However, the adjustment method seems rather ad hoc. I don't quite understand equation (4) in the technical appendix. What is the dependent variable exactly and how is the adjustment applied to the unadjusted prices? I am also concerned with the assumption that value-added share of farm products is assumed to be the same across all countries. The price transmission between sugar and SSB prices depends on the market structures and the cost structure of the soft drink industry. It is therefore unlikely that a country like India and Brazil, net exporters of sugar, would have the same value-added share of farm product as the US. A potential robustness check is to use prices of branded soft drinks (e.g. Coca-Cola) across countries. 2. What does the γ_i, $i=0,1,\dots,5$ mean in equation (2)?
--

VERSION 2 – AUTHOR RESPONSE

Reviewer: 1

Reviewer Name: Lennert Veerman

Institution and Country: Griffith University, Australia

Please state any competing interests or state 'None declared': None declared.

- 1. The authors have clearly done their best to improve the paper on the basis of reviewer comments. They have improved the analysis by adjusting for the fact that sugar price is only one determinant of the overall consumer price of SSBs – though I do wonder if doing so using information on the value-added share of farm products in U.S. food and beverage production is appropriate for these global data.**

The U.S. was only used as a data point to calibrate parameter values for the adjustment equation. Based on equation (4), the value-added is actually an increasing function of income. For countries at the lowest income level, the adjustment is close to 1, so the value-added is close to zero. At this extreme, the sugar price index value would remain unchanged ($adjusted\ price = unadjusted\ price\ index \times 1$) indicating no value added.

For an upper-middle income country (e.g., GDP per capita = \$20,000), the adjustment is 6.4 implying a higher value-added ($adjusted\ price\ index = price\ index \times 6.4$).

Take Switzerland, which is in the highest income decile, $Y = \$50,963$ and the *unadjusted* sugar-price index = 0.63. At this income level, the adjustment factor is 12.39 and the *adjusted* sugar-price index = 7.81.

Examples in the supplemental appendix should make this point clearer to readers.

2. ***I also think the decision to sacrifice the estimates of cross-price elasticities was a good one.***
3. ***But even with the above adjustment, the price elasticity estimates are still far lower than almost any other study. (The highest results for US age groups are about half the size of the lowest findings from several other US studies, as the discussion makes clear.) While it is certainly true that those other studies also have their flaws, I still think a study based on data from around the world in a single year that uses adjusted proxy data for its independent variable is unlikely to provide better estimates. And that is for the US, where the adjustment factor was from. My interpretation is that the study shows that after adjustments, sugar prices are correlated with the intake of SSBs, and sheds some light on which age groups and countries are likely to be more responsive to changes in the (relative) price of sugar, but says little about the absolute magnitude of those responses.***

In the discussion, we do highlight differences as well as similarities with previous findings. We also provide discussion using past studies to show that elasticities based on intakes will be smaller than elasticities based on expenditures.

As we mentioned in the last reply, global analyses often rely on cross-country estimates from cross-sectional analysis, but your point is well taken.

4. ***The questions, then, are whether this study adds enough to the literature to warrant publication, and whether the findings have been adequately put into context by the discussion. I am not sure about the first point. On the second point, the authors do acknowledge weaknesses, and show (as per above) that their estimates of own-price elasticities are much below those of other studies. The discussion does not mention any of these caveats, however.***

In the discussion we do highlight differences as well as similarities with previous findings. We also provide discussion about studies that suggest that elasticities based on intake will be smaller than elasticities based on expenditures.

5. ***Whether these PEs are sufficiently strong to take the next step and estimate the likely impact of SSB taxes is moot. I don't think so; I would keep it at observing that the young and those in low-income countries are likely to be most responsive to such taxes.***

We prefer to keep this information in the paper. While we understand the point being made, the tax projections are another way of explaining the elasticity results and the tax results will likely be clearer to readers unfamiliar with elasticities. The sophisticated reader can always appeal to our upper-bound responses as more "reasonable" results if they are so inclined.

While some of our projections are small (or even insignificant), particularly for middle-aged adults in higher-income countries, this is not implausible. Keep in mind, there are studies that show taxes to be insignificant.

For instance,...

Silver LD, et al. Changes in prices, sales, consumer spending, and beverage consumption one year after a tax on sugar-sweetened beverages in Berkeley, California, US: A before-and-after study. *PLoS Med.* 2017;14(4):e1002283.

...found significant reductions in expenditure but that reductions in daily intake were insignificant.

6. ***PS: While the Estonian government proposed a tax on SSBs that would have come into force by now, and Parliament voted in favour, the country still does not have an SSB tax. In a very unusual move, the President refused to sign, questioning the constitutionality of the exemption for SSBs sold on the ferries to and from the country. The tax was later deemed not to violate the constitution, but the timetable no longer tenable. With that, the matter***

rested with the Parliament again, but with elections in sight the government parties chose not to pick another fight with a powerful vested interest. So there the matter rests.

Thank you for this information and we have deleted Estonia from the text.

Reviewer: 3

Reviewer Name: Cherry Law

Institution and Country: London School of Hygiene and Tropical Medicine

Please state any competing interests or state 'None declared': None declared

While the authors have greatly improved the paper, there are some issues remained:

1. *I appreciated the efforts the authors made to adjust the prices of sugar and fruits for production cost difference across countries. However, the adjustment method seems rather ad hoc. I don't quite understand equation (4) in the technical appendix.*

a) *What is the dependent variable exactly and how is the adjustment applied to the unadjusted prices?*

The adjustment is a multiplicative factor. For instance, if equation (4) is equal to 1.5, then we multiply the price index value by 1.5 to get an *adjusted* price index value.

We now provide more explanation and examples in the supplemental appendix to make this more obvious to readers.

b) *I am also concerned with the assumption that value-added share of farm products is assumed to be the same across all countries.*

In fact, the opposite is true. The value-added is actually an increasing function of national income. For countries at the lowest income level, the adjustment is close to 1, so the value-added is close to zero. At the extreme case (zero income), the sugar price index value would remained unchanged ($adjusted\ price = unadjusted\ price\ index \times 1$) indicating no value added.

For an upper-middle income country (e.g., GDP per capita = \$20,000), the adjustment is 6.4 implying a higher value-added ($adjusted\ price\ index = unadjusted\ price\ index \times 6.4$).

Examples in the supplemental appendix should make this point clearer to readers.

c) *The price transmission between sugar and SSB prices depends on the market structures and the cost structure of the soft drink industry. It is therefore unlikely that a country like India and Brazil, net exporters of sugar, would have the same value-added share of farm product as the US. A potential robustness check is to use prices of branded soft drinks (e.g. Coca-Cola) across countries.*

See our reply to b. The adjustment factor is not the same across countries.

2. *What does the 'i' i=0,1,....,5 mean in equation (2)?*

Thanks for this catch. We mistakenly used the symbol "i" for two different meaning. We change this symbol in equation (2) to "k" since we already used "i" and "j" to denote the intake categories. It should now be clear that "k" is just the parameter subscript.

VERSION 3 – REVIEW

REVIEWER	Lennert Veerman Griffith University, Australia
REVIEW RETURNED	25-May-2019

GENERAL COMMENTS	The authors have defended their approach but did not make any changes. That leads me back to my original assessment: “In essence, in epidemiological terms, this is an ecological study. It tries to predict SSB consumption based on various data from 164 countries around the world at a single point in time. I don’t think this is a very strong design to answer this type of question.” Moreover, the variation in effects by age and sex are based only on variations in intake. The study would be stronger if it used information on consumption and prices in a range of years – analogous to the GBD approach to epidemiology with DisMod meta-regression. Furthermore, instead of consumer prices for sugared drinks and fruit juice, prices for sugar and fruit were used, with an adjustment that is based entirely on one US estimate, supplemented with assumptions. As it is, this is a typical econometric study that tries to make up for shoddy data with very complex modelling and the liberal use of assumptions. This results in estimates that are way outside the range of all previous studies that estimated the same quantities – although one can always find underpowered studies that show insignificant results, as the authors duly did. My advice therefore remains to reject this paper as not scientifically valid. It is up to the editor to determine if this should nevertheless be considered a worthwhile addition to the literature.
--

REVIEWER	Cherry Law London School of Hygiene and Tropical Medicine
REVIEW RETURNED	22-May-2019

GENERAL COMMENTS	While I remain reserved about the price adjustment index, the authors have now highlighted that as a limitation. Overall, they addressed my previous comments satisfactorily.
---

VERSION 3 – AUTHOR RESPONSE

We have confirmed our right to reproduce the map in the supplement. We contacted Esri and confirmed the use of their software for publication. We were provided with the statement below, which has been added to supp. fig 1.

Map was created using ArcGIS® and ArcMap™ software by Esri. ArcGIS® and ArcMap™ are the intellectual property of Esri and are used herein under license. According to Esri citation guidelines, geodata is redistributable with value-added software applications developed by Esri on a royalty-free basis with proper metadata and copyright attribution to the respective data vendor/vendors. See Esri citation guide for more details:

<http://help.arcgis.com/en/arcgisdesktop/10.0/help/index.html#/001z00000003000000.htm>.